# Hypothyroidism Is Correlated with Ventilator Complications and Longer Hospital Days after Coronary Artery Bypass Grafting Surgery in a Relatively Young Population: A Nationwide, Population-Based Study

**DOI:** 10.3390/jcm11133881

**Published:** 2022-07-04

**Authors:** Jiun-Yu Lin, Pei-Chi Kao, Yi-Ting Tsai, Chi-Hsiang Chung, Wu-Chien Chien, Chih-Yuan Lin, Chieh-Hua Lu, Chien-Sung Tsai

**Affiliations:** 1Division of Cardiovascular Surgery, Department of Surgery, Tri-Service General Hospital, National Defense Medical Center, Taipei 11490, Taiwan; acidyulin@gmail.com (J.-Y.L.); cvsallen@ndmctsgh.edu.tw (Y.-T.T.); 2School of Public Health, University of Queensland, Brisbane, QLD 4072, Australia; kaopeichi@hotmail.com; 3Department of Medical Research, Tri-Service General Hospital, National Defense Medical Center, Taipei 11490, Taiwan; g694810042@gmail.com (C.-H.C.); chienwu@mail.ndmctsgh.edu.tw (W.-C.C.); 4School of Public Health, National Defense Medical Center, Taipei 11490, Taiwan; 5Division of Endocrinology and Metabolism, Department of Internal Medicine, Tri-Service General Hospital, National Defense Medical Center, Taipei 11490, Taiwan

**Keywords:** hypothyroidism, coronary artery bypass surgery, ventilator complication, National Health Insurance Research Database

## Abstract

Background: Some research indicated that hypothyroidism has huge adverse effects for the metabolic, cardiovascular, respiratory, and immune systems. However, there is no confirmed conclusion for the effect of cardiovascular surgery. This cohort study aims to investigate the prognosis of hypothyroidism patient at the age under 65-year-old after coronary artery bypass grafting (CABG) surgery. Method: From the National Health Insurance Research Database of Taiwan, 1586 patients with hypothyroidism who underwent elective CABG surgery were selected, along with 6334 patients who underwent surgery in a ratio of 1:4 sex-, age- and index year-matched controls, who were out of hypothyroidism. We used Cox proportional hazard analysis to compare the rate of 30-day, 5-year mortality, post-operative atrial fibrillation, respiratory complication during an average of 10-year follow-up. Result: Post-CABG patients had more hospital days, which was associated with hypothyroidism, male, DM and higher CCI_R (*p* < 0.001). Post-CABG patients had more inpatient respiratory complications, which was associated with hypothyroidism (*p* = 0.041), DM and CCI_R (*p* < 0.001, *p* = 0.046), and there was no difference in 1-year respiratory complication, tracheostomy in the same hospital course and within 1 year, repeated PCI, Af, CVVH, cerebral infarction, 30-day and 5-year mortality rate. Conclusions: Hypothyroidism correlates to post-CABG ventilator-related complications and pneumonia, and prolonged hospital days, but no effect on 30-day, 5-year mortality, post-operative atrial fibrillation and cerebral infarction rate. Thyroid function survey might include routinely preoperative survey for CABG outcome prognosis.

## 1. Introduction

Research indicates that hypothyroidism decreases heart contractility, reduces stroke volume and rate, affects the vascular endothelium, and increases the risk of atherosclerosis, systemic vascular resistance, hypertension [1]. atherogenic lipid profile, and coagulation abnormality [2]. Severe thyroid dysfunction is related to muscle relaxation caused by respiratory muscle depression and sleep apnea syndrome. Surgical stress might be followed by a prolonged recovery process and cardiac dysfunction [3].

It is also reported that impaired renal function is linked to both overt and subclinical hypothyroidism, which are associated with hypotension, increased vascular resistance, and decreased sodium reabsorption, which leads to volume contraction, reduced renal blood flow, and low glomerular filtration rate [4]. 

Hypothyroidism was reported to be strongly related to cardiovascular disease, respiratory complications, and a significant difference in ventilator weaning time [3]. Patients with advanced coronary artery disease (CAD) require a coronary artery bypass grafting (CABG). Some single-center retrospective clinical studies on the long-term prognosis of patients with hypothyroidism who underwent CABG have revealed a higher risk of revascularization, atrial fibrillation, and all-cause mortality [5,6,7] and increased risk of renal replacement therapy [4]. The exact physiology of thyroid disease and the relationship with post-CABG prognosis remain unclear. Thus far, large-scale clinical statistical analysis evidence on the complication rate and long-term mortality is lacking. Current important prognostic assessment tools such as EuroSCORE II [8], and Society of Thoracic Surgeons (STS) score [9] do not include thyroid function assessment, despite the fact that it has a significant impact on the metabolic, cardiovascular, and circulation system.

This cohort study, which used data from the Taiwan National Health Insurance Database (NHIRD), aimed to analyze the correlation between hypothyroidism and post-CABG prognosis including mechanical ventilator supporting time, continuous veno-venous hemofiltration (CVVH), revascularization via percutaneous coronary intervention (PCI), rate of postoperative atrial fibrillation (Af) and cerebral infarction, and 30-day and 5-year all-cause mortality. This cohort study may establish screening, prognosis evaluation, and treatment strategies for patients with perioperative hypothyroidism following CABG.

## 2. Materials and Methods

### 2.1. Database

Taiwan’s National Health Insurance Program has been in operation since 1995 and covers nearly all residents of Taiwan (21,869,478 beneficiaries of 22,520,776 residents at the end of 2002). Over 99% of Taiwan residents have participated in the system. Currently, the NHIRD of the National Institutes of Health in Taiwan is responsible for the complete national health insurance claims database and has released dozens of extracted data sets for researchers. The data set consists of 1,000,000 people in 2000 and collects all records of these people since 1995. These random samples have been confirmed by the National Institutes of Health as representative of Taiwan’s population. These include patient demographics, major and minor diagnoses, procedures, prescriptions, and medical expenditures. It also records all reimbursements for inpatient and outpatient medical services. The NHIRD uses the International Classification of Disease 9th Revision, Clinical Modification (ICD-9-CM), and 10th (ICD-10) codes to record diagnoses. All diagnoses of hypothyroidism, ventilator pneumonia, and complication following CABG were made by a board-certified medical specialist with sufficient reliability and validity. In this cohort data, the original identification number of each patient is encrypted to protect privacy. As the encryption procedures are consistent, claims belonging to the same patient are feasible within the NHIRD. The database is accessible to the public. To protect personal privacy, the electronic database was decoded, and patient identification information was encrypted for the public to further access and research.

This study was exempted from full review by the institutional review board. This study was also evaluated and approved by the National Institutes of Health in Taiwan [10,11,12].

### 2.2. Study Design

This study followed a retrospective, paired cohort design. For patients diagnosed with hypothyroidism ICD-9 (243–244) and ICD-10 (E02, E03–03.3, E03.8–03.9, E89.0), the diagnosis of primary hypothyroidism relies heavily on laboratory tests for high serum thyroid-stimulating hormone (TSH) concentrations >4.5 mIU/L and low serum free thyroxine (T4) <0.7 ng/dL [13]. Each patient must match the serological hormone test as the basis for diagnosis. A certified physician then codes the ICD diagnostic after confirming the laboratory data. The patients who underwent CABG (68023B-68025B) from 1 January 2000, to 31 December 2019 were retrospectively analyzed. The ICD-10 coding started with 2017 records; thus, we compared ICD-9 and ICD-10. Each enrolled patient had a confirmed diagnosis of hypothyroidism and underwent CABG; both cohorts excluded individuals aged >65 years, emergent surgery, re-do operation, and combined valve surgery.

The covariates include the Charlson Comorbidity index (CCI). CCI was first developed in 1987 by Mary Charlson and colleagues [14] as a weighted index to predict risk of death with 1 year of hospitalization for patient with specific comorbid conditions. Nineteen conditions were included in the index, each condition was assigned a weight from 1 to 6, based on the estimated 1-year mortality hazard ratio from a Cox proportional hazards mode. In 1993, Richard Deyo et al. separately adapted the CCI to ICD-9-CM diagnosis that the index could be calculated using administrative data [15]. On the other hand, CCI can be a tool to represent the severity of underlying disease and condition, absence of chronic obstructive pulmonary disease (COPD), sex, age, geographic area of residence (northern, central, southern, and eastern Taiwan), and level of urbanization (levels 1–4). The level of urbanization was determined according to the population size and development level.

### 2.3. Outcome Measures

All study participants were followed from the index date until the onset of ventilator complication, mortality, and all the aforementioned above from the NHI program before the end of 2019.

### 2.4. Statistical Analysis

All statistical analyses were performed using IBM SPSS Statistics for Windows, version 22 (IBM Corp., Armonk, NY, USA). The Chi-square test and *t*-test were used to evaluate the distributions of categorical and continuous variables, respectively. Multivariate Cox proportional hazards regression analysis was used to determine the risk of mortality or hospitalization among patients with CAD complicated with hypothyroidism who underwent CABG. Results of the statistical analyses were presented as hazard ratios (HRs) with 95% confidence intervals (CIs). Differences in the risk of hospitalization or mortality between the groups with and without hypothyroidism were determined using a two-tailed test with a *p* value < 0.05.

### 2.5. Ethics

Our research was conducted in accordance with the ethical code of the World Medical Association (Declaration of Helsinki). The Institutional Review Board of the Tri-Service General Hospital (TSGH) approved our study and waived the need for individual written informed consent (TSGH IRB No. E202216017).

## 3. Results

A total of 1,949,101 patients were enrolled in this study, and 14,121 patients underwent CABG surgery. From these patients, we excluded 3972 who underwent re-do surgery, combined valve surgery, and emergent CABG, were lost to follow-up, and aged >65 years. The study population of 10,149 patients was divided into 1586 patients with hypothyroidism (study group), 8563 patients without hypothyroidism (comparison group) for a four-fold propensity-score matching by sex, age, and index year, and 6334 patients as the control cohort (Figure 1). The follow-up endpoints were 30-day and 5-year mortality, CVVH use, inpatient hospitalization, respiratory complications in 1 year, tracheostomy in 1 year, and PCI until 31 December 2019.

Table 1 presents the baseline characteristics of the study population, including sex, age, comorbidities, geographical location, urbanization, level of care, and income. Of the 7930 patients who underwent CABG, 3505 were male (44.2%), and the mean age was 42.2 ± 21.0 years; no significant differences were noted in age and sex. At baseline, patients who underwent CABG have underlying hypothyroidism, with significantly higher comorbidity with diabetes mellitus (DM) and higher CCI ratio (CCI_R) than those without hypothyroidism.

Table 2 shows the characteristics of the study population at the endpoint. The duration of hospitalization (days), presence of ventilator-acquired pneumonia, complication risk, comorbidity with DM, and CCI_R were significantly higher in the group with hypothyroidism than in the group without hypothyroidism (*p <* 0.001, *p* = 0.002, *p <* 0.001). Figure 2 shows the Kaplan–Meier analysis for the cumulative risk of ventilator complications in the patient and control groups, and the difference was significant (log-rank, *p* = 0.012).

According to the linear regression and Cox regression analysis (Table 3), patients who underwent CABG had a longer hospitalization, which was associated with hypothyroidism, male sex, DM, and higher CCI_R (*p* < 0.001). Patients who underwent CABG had a higher rate of inpatient respiratory complications, which were associated with hypothyroidism (*p* = 0.041), DM (*p* < 0.001), and CCI_R (*p* = 0.046). In addition, patients who underwent CABG with COPD as a comorbidity had a higher rate of requiring tracheostomy during hospitalization and within 1 year of follow-up (*p* < 0.001, *p* = 0.013). The Cox regression analysis also revealed that in the group with hypothyroidism, no difference was noted in the rates of 1-year respiratory complication, tracheostomy during hospitalization and within 1 year of follow-up, repeated PCI, Af, CVVH, cerebral infarction, and 30-day and 5-year mortality.

Table 4 shows that patients who underwent CABG with underlying of hypothyroidism had a higher respiratory complication rate (adjusted HR, 1.507, *p* = 0.041) and prolonged hospitalization, but no effects were noted on the occurrence of 30-day and 5-year mortality, postoperative Af, and cerebral infarction rate.

As regards the insured premium range among patients with hypothyroidism who underwent CABG, a significant difference was found between those with a monthly insured salary of more than TWD 35,000 (USD 1250 USD) and those with less than TWD 18,000 (USD 640) (*p* < 0.05, *p* = 0.031).

## 4. Discussion

To the best of our knowledge, this is the first nationwide population cohort study that analyzed post-CABG outcomes of a relatively young population with hypothyroidism, who were found to have a significantly increased risk of ventilator-acquired pneumonia and complication, and prolonged hospitalization compared with those without hypothyroidism. Although the CCI did not include evaluation of hypothyroidism, comorbidities were more severe in the study groups than in the control groups.

Hypothyroidism adversely affects the immune system, predisposing patients to infections that affect recovery after surgery [16]. Some previous studies obtained similar results and followed the course of our discussion. In a study among patients who underwent orthopedic surgery, the hypothyroidism group had a significantly higher risk of postoperative pneumonia than the control group [17]. Another cohort study reported that patients with hypothyroidism were at a higher risk of heart failure following diagnosis of community-acquired pneumonia than those without hypothyroidism [18].

A possible mechanism for this association is related to the low triiodothyronine (T3) syndrome. In 1990s. Broderick TJ and coworkers [19] established the theory of low T3 syndrome in patients undergoing cardiac procedures who have low free T3 levels, and decreased cardiac 5′-monodeiodinase activity, which reduces the peripheral conversion of thyroxine (T4) to T3. The potential causes of decreased thyroid hormone levels during and after cardiopulmonary bypass (CPB) are varied and include hypothermia, reduced peripheral conversion of T4 to T3, hemodilution, non-pulsatile blood flow, the suppressive effect of cytokines and tumor necrosis factor on thyroid function, skin preparations with iodine, and cortisol-induced effects on thyroid-stimulating hormone (TSH) secretion [20], which alters cardiocirculatory physiology, could contribute to this condition. [5]. CPB simulates the effect of euthyroid sick syndrome [21], and the length of bypass time longer than 90 min increase the severity of hypothyroidism [22]. Pre-operative hypothyroidism may aggregate and amplify the low T3 syndrome, which negatively affects mortality rate and cardiac output after CABG [23]. However, postoperative low T3 syndrome was documented to seriously affect cardiac function. Many well-designed trials that focus on T3 supplement for preventing low T3 syndrome indicate no benefit and are not able to reveal a significant clinical improvement affecting the outcomes [24,25,26]. Deficiency of T3 may be pre-existing and long lasting for CAD patients, and short-term exposure to low T3 condition would not alter the cardiomyocyte morphology [23], which might explain the negative result of short-term T3 supplement therapy lack of significant treatment response.

Another possible mechanism is that hypothyroidism affects respiratory muscles and alveoli, which is one of the problems in ventilator weaning of patients with hypothyroidism. Warner et al. [27] described that thyroid hormones are determinants of the metabolic and contractile phenotypes of skeletal muscles, so they modulate the function of mitochondria. According to Darra et al. [28] impaired diffusing capacity is postulated to be due to a reduction in the pulmonary capillary bed, an increase in the permeability of pulmonary capillaries, and an increase in alveolar and capillary wall thickness caused by a deficiency of thyroid hormones. A decrease in the hypoxic and hypercapnic ventilatory drive of the respiratory muscles, weakness, diaphragmatic dysfunction, and skeletal muscle myopathy could be major causes of prolonged ventilator weaning. Huang et al. [16] opined that the risk of pneumonia may reduce with thyronine replacement, and a low serum thyroxine level is an objective marker. Theoretically, there appears to be no sound justification for the routine use of T3 in patients undergoing CABG [25]. However, T3 supplementation should be considered in the case of postoperative respiratory failure, difficult weaning, complicated pneumonia, and low cardiac output syndrome; such a supplementation might be beneficial but requires further investigation.

Patients with hypothyroidism who underwent CABG had significantly prolonged hospitalization. Hypothyroidism causes more comorbidities and complications with huge effects on the metabolic, cardiovascular, and immune systems. Buller et al. [17] reported an increased risk of multiple perioperative complications and higher costs. CPB induces an acute stress response, causing changes in several hormones levels, elevated cortisone and a decrease in TSH level during the 24 h after surgery, increasing cardiovascular risk [29,30]. Hypothyroidism with hypocoagulable state may increase the risk of hemorrhage and therefore necessitate extra blood transfusion, theoretically increasing the risk of transfusion [31] Surgeons should counsel patients on these findings and seek preoperative optimization strategies to reduce these risks and lower costs.

Previous studies have reported increased mortality, Af, and stroke rates [5,6,7], and repeat target-vessel PCI [6] in patients with hypothyroidism who underwent CABG, and their findings were different from our results. These studies are single-center retrospective reviews that confirmed the diagnosis by routine screening of the serum levels of TSH, T3, and T4 and clinical symptoms described in medical records. Symptoms that occurred during hospitalization may not be coded in the ICD code, and currently, no guidelines are available for preoperative thyroid hormone testing to detect thyroid disease. This cohort study with analysis of ICD coding revealed no significant difference in the aforementioned outcomes. However, the sample size of the single-center study is small and the source of patients is limited by regions and specific ethnic groups. This article analyzes the health insurance database, larger samples and spans different cities, regions, and medical institutions of all levels. It is expected to be closer to the real-world situation, which is also one of the characteristics of the health insurance database.

Thyroid dysfunction is increasingly found in patients with DM, with a prevalence of approximately 13.4% [32]. DM may affect thyroid function to a variable extent, and unrecognized thyroid dysfunction not only worsens the metabolic control but also impedes the management of DM. In the linear regression and Cox regression (Table 4), a significant difference was noted in ventilator complication and pneumonia rate (*p <* 0.05). COPD, obesity, and DM are factors that easily affect ventilator complications. With adjustment and Cox regression, DM revealed a negative effect on the comparison of each outcome. Nevertheless, hypothyroidism significantly causes longer hospital days and respiratory complications.

This study has a few limitations. First, this article does not include the analysis of medication and treatment before or after operation status, and the nature of health insurance database is that it does not contain laboratory values and serological information, in contrast to other single medical center research results using internal database serum lab data analysis to evaluate pre-operative thyroid function, which could be more accurate evaluation. Second, the respiratory complications include ventilator-acquired pneumonia and prolonged ventilator support; this statistic analysis combining both outcomes together does not distinguish a difference between them, so the relationship cannot be discussed through this study. Third, many complications or symptoms that might be temporary are only present in medical records that cannot be revealed by ICD code from the NHI database, meaning that the incidence of comorbidities and adverse events may be underestimated. Furthermore, numerus prescription affecting thyroid function such as amiodarone or iodine contrast medium may have interacted in our study. Lastly, a retrospective design may lead to retrospective and selection biases due to the use of ICD codes; these codes are coded very accurately because hospitals’ coding crew put more efforts into the coding process as they are responsible for billing to the NHI [33]. Finally, the statistical methods for this study included associating hypothyroidism with higher ventilator complications and longer hospital stays.

## 5. Conclusions

To our knowledge, this is the first nationwide, population cohort study to demonstrate that hypothyroidism is associated with CABG, ventilator-related complications, pneumonia, and prolonged hospital stay. In patients undergoing CABG, scheduling thyroid function testing may help to predict whether CABG prognosis is related to hypothyroidism. As this is an observational study, further studies are needed to assess the mechanisms underlying the increased risk of complications after CABG in patients with hypothyroidism.

## Figures and Tables

**Figure 1 jcm-11-03881-f001:**
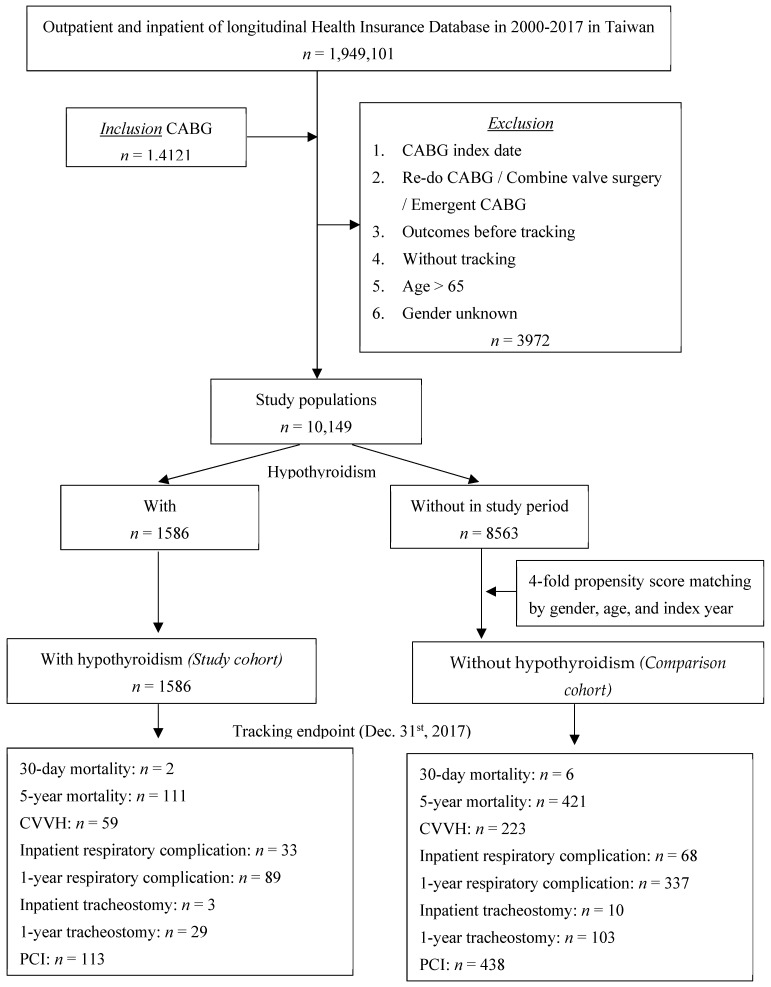
Flow chart of study patient selection from the National Health Insurance Research Database in Taiwan.

**Figure 2 jcm-11-03881-f002:**
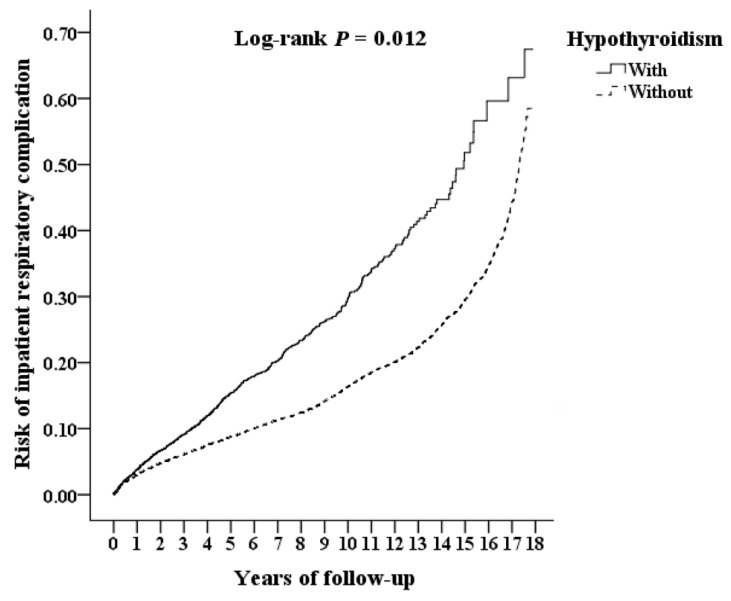
Kaplan–Meier analysis of risk of inpatient respiratory complication among post-CABG patient under aged 65, stratified by diagnosed hypothyroidism and analyzed using a log-rank test.

**Table 1 jcm-11-03881-t001:** Characteristics of study baseline (total 7930).

	Hypothyroidism (1586 20%)		Controls (6344 80%)		*p* Value
	*n*	%	*n*	%	
Gender					0.999
Male	701	44.2	2804	44.2	
Female	885	55.8	3540	55.8	
Age, years Mean±	42.22 ± 21.00		42.18 ± 20.97		0.946
Coexisting medical conditions					
COPD					0.698
without	1473	92.9	5910	93.2	
with	113	7.1	434	6.8	
DM					<0.001
without	1202	75.8	5169	81.5	
with	384	24.2	1175	18.5	
Obesity					0.615
without	1564	98.6	6266	98.8	
with	22	1.4	78	1.2	
CCI_R (Mean ± SD)	1.071.29		0.86 ± 1.17		<0.001
Location					<0.001
Northern	623	39.3	1588	25.0	
Middle	422	26.6	1736	27.4	
Southern	410	25.9	1560	24.6	
Eastern	131	8.3	1460	23.0	
Urbanization level					<0.001
Very high	528	33.3	1914	30.2	
High	603	38.0	2244	35.4	
Moderate	199	12.6	1022	16.1	
Low	256	16.1	1164	18.4	

*p*: Chi-square/Fisher exact test on category variables and t-test on continue variables. COPD: chronic obstructive pulmonary disease. DM: Diabetes mellitus. CCI_R: Charlson comorbidity index.

**Table 2 jcm-11-03881-t002:** Characteristics of study at the endpoint.

Characteristics of Study in the Endpoint							
Hypothyroidism	Total	With			Without		*p*
Variables	*n*	%	*n*	%	*n*	%	
Total	7930		1586	20.00	6344	80.00	
Length of days	19.38 ± 12.33	21.15 ± 13.22			18.94 ± 12.06		<0.001
30-day mortality							0.664
Without	7922	99.90	1584	99.87	6338	99.91	
With	8	0.10	2	0.13	6	0.09	
5-year mortality							0.614
Without	7398	93.29	1475	93.00	5923	93.36	
With	532	6.71	111	7.00	421	6.64	
CVVH							0.941
Without	7638	96.32	1527	96.28	6111	96.33	
With	292	3.68	59	3.72	233	3.67	
Inpatient respiratory complication							0.002
Without	7829	98.73	1553	97.92	6276	98.93	
With	101	1.27	33	2.08	68	1.07	
1-year respiratory complication							0.619
Without	7504	94.63	1497	94.39	6007	94.69	
With	426	5.37	89	5.61	337	5.31	
Inpatient tracheostomy							0.732
Without	7917	99.84	1583	99.81	6334	99.84	
With	13	0.16	3	0.19	10	0.16	
1-year tracheostomy							0.583
Without	7798	98.34	1557	98.17	6241	98.38	
With	132	1.66	29	1.83	103	1.62	
PCI							0.741
Without	7379	93.05	1473	92.88	5906	93.10	
With	551	6.95	113	7.12	438	6.90	
Af							0.380
With	7829	98.73	1562	98.49	6267	98.79	
Without	101	1.27	24	1.51	77	1.21	
Cerebral infarction							0.943
With	7881	99.38	1576	99.37	6305	99.39	
Without	49	0.62	10	0.63	39	0.61	
COPD							0.253
with	7362	92.84	1462	92.18	5990	93.00	
without	568	7.16	124	7.82	444	7.00	
DM							<0.001
with	6317	79.66	1191	75.09	5126	80.80	
without	1613	20.34	395	24.91	1218	19.20	
Obesity							0.326
with	7825	98.68	1561	98.42	6264	98.74	
without	105	1.32	25	1.58	80	1.26	
CCI_R	0.94 ± 1.20		1.08 ± 1.23		0.91 ± 1.19		<0.001

*p*: Chi-square/Fisher exact test on category variables and *t*-test on continue variables.

**Table 3 jcm-11-03881-t003:** Factors of outcomes by using linear regression and Cox regression.

Variables	Log (Length of Days)				30-Day Mortality				5-Year Mortality				CVVH			
	Adjusted RR	95% CI	95% CI	*p*	Adjusted HR	95% CI	95% CI	*p*	Adjusted HR	95% CI	95% CI	*p*	Adjusted HR	95% CI	95% CI	*p*
Hypothyroidism																
Without	Reference				Reference				Reference				Reference			
With	1.289	1.142	1.414	<0.001	1.354	0.854	1.893	0.130	1.073	0.590	1.555	0.504	1.027	0.535	1.609	0.439
Gender																
Male	1.503	1.113	1.797	<0.001	1.035	0.458	1.498	0.528	1.102	0.608	1.810	0.682	1.595	0.901	2.702	0.127
Female	Reference				Reference				Reference				Reference			
Age (years)	1.253	1.118	1.471	<0.001	1.519	1.040	2.004	0.033	1.392	1.111	1.904	<0.001	1.921	1.355	2.330	<0.001
Insured premium (TWD)																
<18,000	Reference				Reference				Reference				Reference			
18,000–34,999	0.874	0.532	1.156	0.286	0.793	0.592	1.012	0.052	0.884	0.495	1.353	0.596	0.773	0.503	0.924	<0.001
≥35,000	0.472	0.283	0.682	<0.001	0.000	-	-	0.999	0.694	0.407	1.037	0.510	0.315	0.188	0.605	<0.001
COPD																
Without	Reference				Reference				Reference				Reference			
With	1.351	0.884	1.706	0.224	1.145	0.775	1.972	0.236	1.101	0.742	1.904	0.275	1.295	0.976	1.562	0.075
DM																
Without	Reference				Reference				Reference				Reference			
With	1.702	1.165	2.801	<0.001	1.501	1.114	1.969	<0.001	1.482	1.085	1.925	0.001	1.797	1.008	2.465	0.041
Obesity																
Without	Reference				Reference				Reference				Reference			
With	1.165	0.898	1.379	0.181	1.189	0.674	1.870	0.331	1.160	0.632	1.814	0.365	1.320	0.775	1.765	0.277
CCI_R	1.325	1.147	1.696	<0.001	1.268	0.910	1.505	0.118	1.322	1.060	1.691	0.004	1.921	1.242	2.814	<0.001
**Variables**	**Inpatient Respiratory Complication**				**1-Year Respiratory Complication**				**Inpatient Tracheostomy**				**1-Year Tracheostomy**			
	**Adjusted HR**	**95% CI**	**95% CI**	** *p* **	**Adjusted HR**	**95% CI**	**95% CI**	** *p* **	**Adjusted HR**	**95% CI**	**95% CI**	** *p* **	**Adjusted HR**	**95% CI**	**95% CI**	** *p* **
Hypothyroidism																
Without	Reference				Reference				Reference				Reference			
With	1.507	1.023	1.894	0.041	1.075	0.648	1.497	0.387	1.217	0.393	2.000	0.806	1.141	0.315	1.921	0.820
Gender																
Male	1.322	0.692	1.725	0.392	1.304	0.687	1.708	0.401	1.313	0.734	1.799	0.278	1.375	0.803	1.834	0.228
Female	Reference				Reference				Reference				Reference			
Age (years)	1.781	1.122	2.154	<0.001	1.802	1.130	2.165	<0.001	1.584	1.098	2.000	0.001	1.620	1.116	2.033	<0.001
Insured premium (TWD)																
<18,000	Reference				Reference				Reference				Reference			
18,000–34,999	0.999	0.782	1.325	0.382	0.967	0.764	1.293	0.408	0.986	0.785	1.322	0.431	0.925	0.654	1.315	0.504
≥35,000	0.914	0.734	1.306	0.361	0.897	0.692	1.262	0.395	0.924	0.762	1.302	0.417	0.878	0.647	1.253	0.496
COPD																
Without	Reference				Reference				Reference				Reference			
With	1.464	0.652	2.201	0.424	1.202	0.532	2.004	0.485	1.562	1.112	2.241	<0.001	1.462	1.077	2.065	0.013
DM																
Without	Reference				Reference				Reference				Reference			
With	1.972	1.189	2.689	<0.001	1.771	1.095	2.448	0.001	1.707	1.229	2.677	<0.001	1.606	1.191	2.571	<0.001
Obesity																
Without	Reference				Reference				Reference				Reference			
With	1.553	0.722	2.374	0.365	1.345	0.506	2.121	0.503	1.423	0.977	1.986	0.078	1.322	0.883	1.875	0.187
CCI_R	1.137	1.020	1.293	0.046	1.196	1.033	1.278	0.034	1.279	1.050	1.620	0.020	1.315	1.087	1.656	0.012
**Variables**	**PCI**				**Af**						**Cerebral Infarction**					
	**Adjusted HR**	**95% CI**	**95% CI**	** *p* **	**Adjusted HR**	**95% CI**	**95% CI**		** *p* **		**Adjusted HR**		**95% CI**	**95% CI**	** *p* **	
Hypothyroidism																
Without	Reference				Reference						Reference					
With	1.049	0.428	1.640	0.518	1.232	0.797	1.796		0.201		1.021		0.642	1.522	0.479	
Gender																
Male	1.137	0.687	1.805	0.403	1.294	0.896	1.652		0.232		1.456		0.785	1.779	0.303	
Female	Reference				Reference						Reference					
Age (years)	1.282	0.884	1.691	0.135	1.154	0.795	1.486		0.332		1.106		0.684	1.335	0.498	
Insured premium (TWD)																
<18,000	Reference				Reference						Reference					
18,000–34,999	0.529	0.319	0.807	<0.001	0.625	0.482	0.868		<0.001		0.523		0.384	0.774	<0.001	
≥35,000	0.493	0.298	0.703	<0.001	0.595	0.444	0.801		<0.001		0.303		0.125	0.552	<0.001	
COPD																
Without	Reference				Reference						Reference					
With	1.253	0.862	1.672	0.266	1.456	0.799	1.776		0.325		1.562		0.335	2.979	0.776	
DM																
Without	Reference				Reference						Reference					
With	1.553	1.080	1.992	<0.001	1.996	1.124	2.984		<0.001		1.785		1.336	2.201	<0.001	
Obesity																
Without	Reference				Reference						Reference					
With	1.456	0.365	2.605	0.775	1.365	0.756	1.896		0.335		1.485		0.598	1.992	0.483	
CCI_R	1.806	1.452	2.274	<0.001	1.711	1.365	2.154		<0.001		1.562		1.184	1.985	<0.001	

**Table 4 jcm-11-03881-t004:** Rate of outcomes.

Hypothyroidism	With			Without (Reference)			With vs. Without (Reference)			
Outcomes	Events	PYs	Rate (per 10^5^ PYs)	Events	PYs	Rate (per 10^5^ PYs)	Adjusted HR	95% CI	95% CI	*p*
30-day mortality	2	15,698.33	12.74	6	63,105.80	9.51	1.354	0.854	1.893	0.130
5-year mortality	111	15,707.12	706.69	421	63,108.25	667.11	1.073	0.590	1.555	0.504
CVVH	59	15,868.22	371.81	233	63,449.11	367.22	1.027	0.535	1.609	0.439
Inpatient respiratory complication	33	15,833.10	208.42	68	63,438.73	107.19	1.507	1.023	1.894	0.041
1-year respiratory complication	89	15,721.94	566.09	337	63,397.40	531.57	1.075	0.648	1.497	0.387
Inpatient tracheostomy	3	15,860.84	18.91	10	63,425.07	15.77	1.217	0.393	2.000	0.806
1-year tracheostomy	29	15,843.36	183.04	103	63,385.62	162.50	1.141	0.315	1.921	0.820
PCI	113	15,702.45	719.63	438	63,121.25	693.90	1.049	0.428	1.640	0.518
Af	24	15,889.94	151.04	77	63,304.11	121.64	1.232	0.797	1.796	0.201
Cerebral infarction	10	15,876.17	62.99	39	63,202.45	61.71	1.021	0.642	1.522	0.479

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
