# Peer review of "Hypothyroidism Is Correlated with Ventilator Complications and Longer Hospital Days after Coronary Artery Bypass Grafting Surgery in a Relatively Young Population: A Nationwide, Population-Based Study"

_jcm, 2022, doi:10.3390/jcm11133881_

Round 1

Reviewer 1 Report

The submitted manuscript analyzes the extent to which the presence of hypothyroidism affects the prognosis and complications of patients who have undergone CABG. It represents a relatively large sample of patients, the scientific question raised is interesting and the hypothesis does not lack biological plausibility - the data certainly deserves to be published.

I have the following problems and objections:

1. The methodology is described relatively briefly and lacks some rather substantial facts. In particular, it is necessary to clearly describe how hypothyroidism was defined. If the presence of hypothyroidism is determined only based on clinical impression or medical history, this should be clearly stated. In this context, I do not understand how citation 10 relates to this issue.

2. Are there any data available on the treatment of hypothyroidism in the evaluated patients? If so, this parameter should be taken into account - if not, it should be declared as a limitation of the analysis.

3. How did the analysis treat patients with hyperthyroidism? Were they excluded (which I would recommend) or taken to a control group?

4. Charlson comorbidity index should be defined/described in Methods for those (like me) who are not familiar with this index

5. There is some data on the concomitant medication used by patients - if not, this fact should be declared as a limitation of the study. A similar major limitation is the fact that we do not know (as I assume) the actual thyroid status of patients at the time of surgery.

Author Response

We are deeply honored by the time and effort you spent in reviewing this manuscript. By reviewing and revising our manuscript, we were spurred to read more and thus learn more from your criticisms.

Reviewer’s comment 1: The methodology is described relatively briefly and lacks some rather substantial facts. In particular, it is necessary to clearly describe how hypothyroidism was defined. If the presence of hypothyroidism is determined only based on clinical impression or medical history, this should be clearly stated. In this context, I do not understand how citation 10 relates to this issue.

Authors’ response: We do appreciate for this important point. The patients enrolled in this study are mainly from the National Health Insurance Research Database (NHIRD) in Taiwan which covers almost all residents in Taiwan since 1995. The diagnosis of hypothyroidism is based on the coding system (ICD-9 and ICD-10) for insurance reimbursement from the certified physicians. The flowchart of this study is summarized in Figure 1. The diagnosis of primary hypothyroidism is based on clinical symptoms and laboratory results, such as elevated total cholesterol level, delayed Achilles reflex time and other symptoms associated with decrease metabolic rate, also relies heavily on laboratory tests for high serum thyroid-stimulating hormone (TSH) concentrations >4.5mIU/L and low serum free thyroxine (T4) concentrations < 0.7ng/dL. Each patient must match the serological hormone test as the basis for diagnosis. A certified physician than codes the ICD diagnostic after confirming the laboratory data. (Page 5, Line 26-28) The reference 10 is deleted from the revised version. Thanks again for the detailed review.

Reviewer’s Comment 2: Are there any data available on the treatment of hypothyroidism in the evaluated patients? If so, this parameter should be taken into account - if not, it should be declared as a limitation of the analysis.

Authors’ Response: Thanks so much for this important perspective. We don’t have the access for the individual laboratory data from the enrolled patients in the database. We definitely agree with you that this is a limitation of our study and has added it been written into the limitation paragraph of the discussion. (Page 12, line 24-27)

Reviewer’s Comment 3: How did the analysis treat patients with hyperthyroidism? Were they excluded (which I would recommend) or taken to a control group?

Authors’ Response: Many thanks for raising this important issue in our manuscript. Since the patients enrolled in this study were obtained from the NHIRD in Taiwan (as the flow chart showed in Figure 1), the patients enrolled in this study are CABG patients with hypothyroidism who are the main focus of this study. The existing research database with hyperthyroidism ICD-10(E05. E05.90) as the result below.

We agree with you that hyperthyroidism patients maybe another patient groups of interest and we will consider to do some investigation on them. Thank you again for pointing this out.

Hypothyroidism

Total

With

Without

n

%

n

%

n

%

Total

7,930

1,586

20.00

6,344

80.00

Hyperthyroidism

Without

7,746

97.68

1,573

99.18

6,173

97.30

With

184

2.32

13

0.82

171

2.70

Reviewer’s Comment 4: Charlson comorbidity index should be defined/described in Methods for those (like me) who are not familiar with this index.

Authors’ Response: We are deeply appreciated for your valuable suggestions. We have elaborated and revised our Materials and Methods based on your comment. We have added some sentence to introduce CCI in the revised manuscript as below. ( Page 6; Line 8-15)

“The Charlson comorbidity Index (CCI) was first developed in 1987 by Mary Charlson and colleagues as a weighted index to predict risk of death with 1 year of hospitalization for patient with specific comorbid conditions. Nineteen conditions were included in the index, each condition was assigned a weight from 1 to 6, based on the estimated 1-year mortality hazard ratio from a Cox proportional hazards mode. In 1993, Richard Deyo et al. separately adapted the CCI to ICD-9-CM diagnosis that the index could be calculated using administrative data in the other hands. CCI can be a tool to represent the severity of underlying disease and condition.”

Reviewer’s Comment 5: There is some data on the concomitant medication used by patients - if not, this fact should be declared as a limitation of the study. A similar major limitation is the fact that we do not know (as I assume) the actual thyroid status of patients at the time of surgery.

Authors’ response: Many thanks for raising this important question. This article does not include medication and treatment in these enrolled patients with hypothyroidism, we have declared it clearly in the limitation as your valuable suggestions and thanks for the understanding of these limitations. (Page 12, Line 24-27)

Reviewer 2 Report

The text spelling has to be controlled. In the discussion you can consider that  during cardio-pulmonary bypass can disrupt/damage thyroid hormone molecules. Is the length of bypass proportionally to the severity of hypothyroidism? There are other hormonal changes, in particular of cortisol? Did you control TSH, T4 and T3 in the early post-operative? There is a correlation between hypothyroidism with intra-operative transfusion? 

Author Response

Response to Reviewer #2

We are deeply honored by the time and effort you spent in reviewing this manuscript. By reviewing and revising our manuscript, we were inspired to read more and thus learn more from your criticisms.

Reviewer’s comment: The text spelling has to be controlled. In the discussion you can consider that during cardio-pulmonary bypass can disrupt/damage thyroid hormone molecules. Is the length of bypass proportionally to the severity of hypothyroidism? Did you control TSH, T4 and T3 in the early post-operative? There is a correlation between hypothyroidism with intra-operative transfusion?

Authors’ response: Thanks for pointing this important issue and the detailed review. The patients enrolled in this study are mainly from the National Health Insurance Research Database (NHIRD) in Taiwan which covers almost all residents in Taiwan since 1995. The diagnosis of hypothyroidism is based on the coding system (ICD-9 and ICD-10) for insurance reimbursement from the certified physicians. The flowchart of this study is summarized in Figure 1. We totally agree with you that there are some limitations from the kind of “dry-lab “studies. To address this important issue, we have added a paragraph in the Discuss section and pointed out all the possible limitations of this study as below: (From page12 ,Line 24 to Page 13 ,Line 10):

“There are several limitations in this study. First, this study does not include the analysis of medication and treatment before or post operative status and the health insurance database is that it does not contain laboratory values and serological information, other single medical center research results using internal database serum lab data analysis to evaluate pre-operative thyroid function, which could be more accurate evaluation. Second, the respiratory complication include ventilator acquired pneumonia and prolong ventilator support, this statistic analysis combines both outcomes together do not distinguish their difference, the relationship cannot be discussed through this study. Third, many complications or symptoms might temporary happened only present in medical record that cannot reveal by ICD code from NHI database, the incidence of comorbidities and adverse events may be underestimated. Furthermore, numerus prescription affecting thyroid function like amiodarone or iodine contrast medium may have interacted in our study. Lastly, a retrospective design may lead to retrospective and selection biases due to the use of ICD codes, these codes are coded very accurately because hospitals’ coding crew put more efforts into the coding process as they are responsible for billing to the NHI (32).”

Thanks again for such a precise comment,

Round 2

Reviewer 2 Report

 The text is ameliorated and more clear.